# Genome-Wide Identification, Expansion, and Evolution Analysis of Homeobox Gene Family Reveals TALE Genes Important for Secondary Cell Wall Biosynthesis in Moso Bamboo (*Phyllostachys edulis*)

**DOI:** 10.3390/ijms23084112

**Published:** 2022-04-08

**Authors:** Feng Que, Qingnan Liu, Ruofei Zha, Aisheng Xiong, Qiang Wei

**Affiliations:** 1Co-Innovation Center for Sustainable Forestry in Southern China, Bamboo Research Institute, Nanjing Forestry University, Nanjing 210037, China; quefeng@njfu.edu.cn (F.Q.); liuqingnan24@163.com (Q.L.); ruofeizha9702@163.com (R.Z.); 2State Key Laboratory of Crop Genetics and Germplasm Enhancement, College of Horticulture, Nanjing Agricultural University, Nanjing 210095, China

**Keywords:** homeobox gene family, TALE gene family, secondary cell wall, syntenies, WGCNA, moso bamboo

## Abstract

The TALE gene family is a subfamily of the homeobox gene family and has been implicated in regulating plant secondary growth. However, reports about the evolutionary history and function of the *TALE* gene family in bamboo are limited. Here, the homeobox gene families of moso bamboo *Olyra latifolia* and *Bonia amplexicaulis* were identified and compared. Many duplication events and obvious expansions were found in the TALE family of woody bamboo. *PhTALE*s were found to have high syntenies with *TALE* genes in rice. Through gene co-expression analysis and quantitative real-time PCR analysis, the candidate *PhTALE*s were thought to be involved in regulating secondary cell wall development of moso bamboo during the fast-growing stage. Among these candidate *PhTALE*s, orthologs of *OsKNAT7*, *OSH15*, and *SH5* in moso bamboo may regulate xylan synthesis by regulating the expression of IRX-like genes. These results suggested that *PhTALE*s may participate in the secondary cell wall deposition in internodes during the fast-growing stage of moso bamboo. The expansion of the TALE gene family may be implicated in the increased lignification of woody bamboo when divergent from herbaceous bamboos.

## 1. Introduction

The homeobox gene family is a class of genes which have a 60 amino acid helix-turn-helix DNA binding motif. The characteristic domain is called a homeodomain (HD) [1]. The homeobox was originally discovered in fruit fly *Drosophila melanogaster* in 1984 [2]. With the development of RNA-seq, many HD-containing genes have been identified across eukaryotes, for instance, *Arabidopsis thaliana*, poplar (*Populus trichocarpa*), sorghum (*Sorghum bicolor*), rice (*Oryza sativa*), maize (*Zea mays* L.), moso bamboo (*Phyllostachys edulis*), and *Selaginella moellendorffii* [2,3].

In plants, homeobox family genes were classified into 14 subclasses based on the HD sequences and the characteristic codomains. The 14 subclasses are as follows: Zinc finger-HD (PLINC), PINTOX, PHD, BLH, NDX (Nodulin homeobox genes), KNOX (KNOTTED1-like homeobox), SAWADEE, HD-leucine zipper (HD-ZIP) I to IV, DDT, WUSCHEL-like homeobox (WOX), and Luminidependens (LD) [2]. Among the 14 subclasses, KNOX and BLH subfamilies are also known as the three-amino-acid-loop-extension (TALE) superclass [4,5]. In addition, KNOX and BLH proteins always form heterodimers when regulating related biology processes [6,7]. In plants, the TALE superclass is thought to be the oldest class among the homeobox gene family. Genes belonging to the TALE class have been widely studied in model plants and found to play important roles in controlling and regulating formation and maintenance of meristem stem, organ morphogenesis, and second cell wall development [2].

Plant cell walls are a complex network composed of cellulose, hemicellulose, pectin, lignin, and some glycoproteins that encase cells. The network provides structural and mechanical support for plants [8]. Plant cell walls are classified into primary cell walls and secondary cell walls. Primary cell walls are thin, expandable, and mainly composed of cellulose, hemicellulose, and pectin. Compared with primary cell walls, secondary cell walls are thicker and composed of cellulose, hemicellulose, and lignin [9]. The secondary cell wall deposition always happens after the cessation of cell growth and only in some special cell types (sclerenchyma fiber cells and vessel elements). During the formation of secondary cell walls, many transcription factors are involved in the regulating network [10]. Some members of TALE family play important roles in the regulating network. In Arabidopsis, KNOTTEDLIKE FROM ARABIDOPSIS THALIANA1 (BP/KNAT1), a class I KNOX protein, can promote the differentiation of xylem in roots by inhibiting the expression of boundary genes (BLADE-ONPETIOLE1/2) and inhibit lignin synthesis by regulating caffeic acid O-methyltransferase (COMT) and caffeoyl coenzyme A 3-*O*-metyltransferase (CCoAoMT) directly. Mutations in BP can significantly repress or eliminate fiber formation in the xylem of Arabidopsis inflorescences [11]. Like BP, AtKNAT7 (class II KNOX protein) is also a negative regulator in Arabidopsis secondary cell wall deposition. *AtKNAT7* will form a heterodimer with *AtBLH6*, which inhibits the lignin synthesis in the secondary cell wall [12,13]. *AtKNAT7* was also found to promote the biosynthesis of xylan [14,15]. Xylan is the major hemicellulosic polysaccharide in plant secondary cell walls. In cotton (*Gossypium hirsutum* L.), overexpression of *GhKNAT7-A03* and *GhBLH6-A13* can also repress the lignocellulose synthesis in Arabidopsis interfascicular fibers [16]. AtKNAT3 is another class II KNOX protein, which was defined as a potential transcriptional activator in regulating secondary cell wall biosynthesis. Overexpression of *AtKNAT3* can increase the wall thickness of an interfascicular fiber cell [17]. In rice, seed shattering is an important agronomical trait. OSH15 is a class I KNOX protein and SH5 is a BELL-like protein, and a heterodimer of the two proteins can enhance seed shattering by repressing lignin synthesis directly [18]. In addition, *OsKNAT7* also plays a role in regulating the biosynthesis of the secondary cell wall, different from *AtKNAT7* in Arabidopsis. *OsKNAT7* overexpressing lines have decreased cellulose content and knat7 mutants have significantly increased cellulose content. NAC31 is a master transcription regulator in the upstream secondary cell wall regulating network. OsKNAT7 can slow down the cell wall thickness during rice internode development by interacting with NAC31 [19]. The regulating mechanism of secondary cell wall biosynthesis is complex and may be distinct among grass species and many dicots. The orthologous genes from different species, such as *OsKNAT7* and *AtKNAT7*, may play different roles in regulating secondary cell wall development [20].

Bamboos, the perennial plants belonging to the Poaceae family, are classified into herbaceous and woody bamboos [21]. The origination of woody bamboos comes from the divergence of herbaceous bamboos about 42 million years ago. Compared with herbaceous bamboos, woody bamboos have a tree-like culm with more lignification and mechanical strength [22]. Bamboos have a wide distribution around the tropical, subtropical, and temperate regions, and natively distribute in all continents except Antarctica and Europe. Among these bamboos, the whole-genome sequencing of moso bamboohas been released. An ancient whole-genome duplication (WGD) event that occurred 7–12 million years ago was thought important in the evolution of moso bamboo [22,23]. Moso bamboo is a typical representative of temperate woody bamboo [23]. It is one of the most fast-growing plants in the world. The highest daily (24 h) increment can be up to 100 cm [24]. The fast elongation growth of bamboo culms is necessary, accompanied by vigorous cellular processes (cell expansion and cell wall reinforcement). Many new polysaccharides and secondary cell wall components are synthesized and deposited to confer rigidity and mechanical strength for the cell walls [25]. Many research works have been performed to study the regulation mechanism of cell wall development. Some key regulators have been found, such as members of NAC, MYB, and HD gene families [26]. Up to now, most related works were performed in model plants, such as rice, *Arabidopsis*, and poplar, but limited work has been performed in bamboos. In this study, the homeobox gene families of one herbaceous bamboo (*O. latifolia*) and two woody bamboos (*P. edulis* and *B. amplexicaulis*) were identified and compared. Then, duplication events, expansion, collinearity, and expression patterns of the homeobox gene family in moso bamboo were further analyzed. This study provides basal information for further research on the homeobox family members in bamboos.

## 2. Results

### 2.1. Identification and Classification of Homeobox Genes in Bamboos

To analyze the homeobox gene family in bamboos, HMMER 3.0 was used to search homeobox genes against local bamboo genome databases. Initially, 215, 255, and 106 candidate homeobox genes were detected from *P. edulis*, *B. amplexicaulis*, and *O. latifolia*, respectively. Then, to determine the members of homeobox families in moso bamboo, Pfam and SMART databases were adopted to analyze the domain structures of the candidate proteins. As a result, 206, 246, and 105 homeobox genes were determined in *P. edulis*, *B. amplexicaulis*, and *O. latifolia*, respectively.

The homeobox gene family can be classified into 14 subclasses based on the presence of the HD domain and some other codomains. Here, homeobox gene families of *P. edulis*, *B. amplexicaulis*, and *O. latifolia* were classified based on the domain structure analysis through the Pfam database and the phylogenetic analysis (Figure 1, Appendix A). The homeobox gene family in moso bamboo was classified into 12 subclasses, no gene was classified into the NDX and LD subclasses, and 1 gene (*PH02Gene32607.t1*) was unclassified. The quantity of homeobox genes in different plant species varies greatly. Here, the homeobox gene family of some model plants was collected and compared. In monocots, obvious expansion appeared in PLINC, KNOX, and BLH subclasses of moso bamboo compared with homeobox gene families in rice and maize. In moso bamboo, 29, 24, and 31 genes were identified in PLINC, KNOX, and BLH subclasses, respectively. The quantities of the three subclasses are twice as much as that in the rice and maize homeobox gene families. Compared with herbaceous bamboo, obvious expansion also appeared in moso bamboo, especially in PLINC, KNOX, and BLH subfamilies. In KNOX and BLH subfamilies, the quantity in moso bamboo is twice as much as in *O. latifolia*. In another woody bamboo (*B. amplexicaulis*), obvious expansion also appeared in PLINC, KNOX, and BLH subclasses compared with herbaceous bamboo. KNOX and BLH subclasses are together known as the TALE subclass. According to previous reports, some *TALE* genes play important roles in regulating secondary cell wall development. As a result, the TALE subclass attracts us to study whether the expansion of the TALE family related to the divergence of woody bamboo from herbaceous. Members of the TALE gene family were numbered following Appendix A.

### 2.2. Expansion and Evolution Events of Homeobox Gene Family in Moso Bamboo

Gene duplication events are an important way for plants to produce genetic novelty. Compared with some model plants (such as rice, Arabidopsis, and poplar), obvious expansion happened in the moso bamboo homeobox gene family (Table 1). WGD and single-gene duplication are two main duplication types. In addition, single-gene duplication models include four types, tandem duplication (TD), dispersed gene duplication (DD), proximal duplication (PD), and DNA-transposed duplication (TSD). Herein, a software named DupGen_finder was used to analyze the duplication types of homeobox gene members in moso bamboo (Figure 2A). As a result, 186, 7, 6, 2, and 4 genes were identified as WGD (91%), TSD (3%), DD (3%), PD (1%), and TD (2%) type (Figure 2B), respectively. Then, a total of 238 duplication gene pairs were found in the moso bamboo homeobox gene family. Among the 238 gene pairs, 213 were WGD pairs, 12 were DD pairs, 2 were PD pairs, 4 were TD pairs, and 7 were TSD pairs. All these gene pairs distributed within 23 chromosomes of moso bamboo, except chromosome 1 (Figure 2A). In PD, gene pairs are distributed within chromosomes 8 and 21. TD gene pairs are distributed within chromosomes 8, 13, 15, and 17 (Figure 2A).

To study the selection pressure type that acts on moso bamboo homeobox genes, the *Ka*, *Ks*, and *Ka*/*Ks* values were computed for each gene pair. Different gene duplication modes exhibited divergent *Ka* and *Ks* distributions (Appendix A). DSD- and PD-derived gene pairs had higher *Ka*/*Ks* values than other modes (Figure 2C), which indicated that these genes have faced stronger selection pressures. The homeobox gene family’s *Ka*/*Ks* value distribution is similar to that of the moso bamboo whole genome. Notably, *Ka*/*Ks* values of all the gene pairs are less than 1, implying that purify selection plays the main role during the evolution of moso bamboo. In the moso bamboo homeobox gene family, three duplication types (WGD, TD, PD) were identified (Figure 2D). The WGD duplication mode makes major contributions in the expansion of homeobox gene family of moso bamboo.

### 2.3. Synteny Analysis of Homeobox Genes in Moso Bamboo and Rice

As an important crop plant, many research works have been performed on the growth and genetics of rice. As a result, many functioning rice genes have been identified and annotated. Moso bamboo and rice are members of Poaceae family, having adjacent genetic relationships. Therefore, rice is an important reference plant in bamboo research. Herein, we performed collinearity analyses of the homeobox gene family for moso bamboo and rice (Figure 3A). Loss and expansion of homeobox genes were discovered in moso bamboo when compared with rice. A total of 90 rice homeobox genes were found to have a collinear relationship with moso bamboo homeobox genes. The longest collinear segment (Ph15-Os3 and Ph21-Os3) between rice and moso bamboo contained 15 orthologous gene pairs. In addition, the collinear segment between Ph14 and Os1 contains 12 orthologous gene pairs. However, Ph1 has no collinear segment among the rice chromosomes. Most rice homeobox genes showed one-to-two (43) and one-to-four (29) corresponding relationships with homeobox genes in moso bamboo.

Meanwhile, 174 (85%) moso bamboo homeobox genes were found having orthologous genes in the rice genome. To further analyze the role of *TALE* genes in internode development of moso bamboo, the collinear relationship of the TALE subclass between rice and moso bamboo was analyzed in detail (Figure 3B and Appendix A). A total of 21 (81%) *OsTALE* genes have orthologous genes in moso bamboo (Appendix A). Among the 21 *OsTALE* genes, 12 have two orthologous genes, 4 have four orthologous genes, 2 have three orthologous genes and only 3 have one orthologous gene in moso bamboo. The main collinear segments distributed on the chromosome pairs of Ph15-Os3, Ph21-Os3, and Ph5-Os3, respectively. The three collinear segments contained eight, seven, and five orthologous gene pairs, respectively. These results indicated that the TALE subclass of moso bamboo expanded obviously during the evolution and had high syntenies with TALE genes of rice. 

### 2.4. WGCNA Analysis of TALE Genes in Moso Bamboo

To study the potential function of *PhTALE* genes, WGCNA was adopted to construct a co-expression network. Herein, two networks were constructed by using two data sets from different projects which were released on the NCBI. The two projects focused on the development of bamboo shoots and internode fast growth in moso bamboo, respectively. Then, *PhTALE* genes were found clustered in blue and brown modules in the two networks, respectively. In addition, 21 and 13 *PhTALE* genes were found in brown and blue modules, respectively (Figure 4A,C). To know which biology processes these genes participated in, gene ontology (GO) enrichment analysis was performed for brown and blue modules. As a result, most genes of the two modules were enriched in the cell-wall-related biology processes in biology process (BP) ontology. Furthermore, genes in the brown module were found to enrich the biology processes related to secondary cell wall development, such as the lignin biosynthetic process, lignin metabolic process, plant-type secondary cell wall biogenesis, and regulation of secondary cell wall biogenesis (Figure 4B), while in the blue module, genes were found enriching cell wall organization or biogenesis, as well as carbohydrate and polysaccharide biosynthetic processes (Figure 4D). To further study which stage the genes from different modules participating in cell wall development are in, we searched and analyzed all the genes related to the cell wall from the two modules. A total of 87 genes were found in the brown module. Among the 87 genes, 35 genes were annotated as lignin-related genes and 15 were cellulose-related (Figure 5A). In the blue module, 104 cell-wall-related genes were found. Among the 104 genes, 24 were pectin-related, 21 were cellulose-related (Figure 5B). However, *PhTALE* genes that clustered in the blue and brown modules are very different. The two modules only shared four *PhTALE* (*PhBLH6*, *PhBLH9*, *PhKNOX4*, *PhKNOX5*) genes with each other. The young bamboo shoot is the initiation of stem elongation which is different from the rapid growing internode in cellular processes. Co-expression networks were constructed with data from two different growth stages, which may partly explain the difference between the blue and brown modules. All these results suggested that genes clustered in the brown and blue modules may be involved in the regulating network of the cell wall development of moso bamboo.

### 2.5. Expression Profile of Candidate PhTALE Genes

According to the results of the WGCNA analysis, some members of the *PhTALE* subclass were found to participate in regulating cell wall development in moso bamboo. To further study what roles these *PhTALE* gene playing in regulating cell wall development, the expression profiles of these candidate *PhTALE* genes were analyzed. The expression profile of candidate *PhTALE* genes were analyzed with the publicly available sequence data from NCBI (PRJNA342231 and PRJNA547876). One project (PRJNA342231) sequenced different parts of young moso bamboo shoots (approximately 25 cm above ground height), including shoot apical meristem (SAM), young internode (YIN), young node (YNO), basal mature internode (MIN), and basal mature node (MNO). Another project (PRJNA547876) studied the transcriptome in different growth stages of moso bamboo. In the project, three growth stages (starting of cell division (SD), rapid division (RD), and rapid elongation (RE)) were determined based on histological and biochemical analyses in moso bamboo shoots. Interestingly, all the candidate *PhTALE* genes except *PhKNOX13* had significant high expression in the rapid RE growth stage (Figure 6A). *PhKNOX13* showed significant high expression in the SD growth stage. According to the report, cell division is the main cellular process in SD, while cell elongation is the main cellular process in RE. Compared with SD, cell walls in the RE growth stage had higher lignin and cellulose contents. Among different parts of young moso shoots, most *PhTALE* genes showed high expression in mature basal tissues (MNO and MIN). Only *PhBLH11*, *PhBLH3*, *PhKNOX1*, *PhKNOX3*, *PhKNOX13*, *PhBLH10*, *PhBLH16*, *PhKNOX10*, *PhKNOX12*, and *PhKNOX4* had high expression levels in young tissues (SAM, YIN, and YNO) (Figure 6B). Compared with young tissues, lignin biosynthesis is more active in mature tissues. The expression profile of lignin- and cellulose-related genes in the RE growth stage and MNO also suggest that cell walls in RE and MNO had higher lignification (Appendix A).

### 2.6. Expression Profile of Candidate PhTALE Genes in Moso Bamboo Internode during Fast-Growing Stage

The fast growth of bamboo shoots is an interesting physiological characteristic of bamboo plants. The fast elongation of the whole bamboo culm is supported by the elongation growth of every single internode. In a single internode, cells in the upper part usually mature first, then the middle part. The basal part is usually considered as a cell division region. Here, internodes with 15–17 cm length from ~3.5 m high moso bamboo shoots were chosen for further study. The upper, middle, and basal parts of every internode were collected as experimental materials (Appendix A). Firstly, the color of the internodes was found gradually darker from bottom to top (Appendix A). Then, toluidine blue was used to stain the sections of different internode parts. Lignified cell walls are dyed blue when stained with toluidine blue. Herein, cell walls of the vascular bundle outer sheath in the upper part were obviously dyed blue (Figure 7A–F), while in the middle and basal parts, the blue color was less, and not obvious. These results suggested that the upper part had higher lignification than the middle and basal parts of the internode. The contents of cellulose and lignin in the three parts also supported this conclusion (Figure 7D,H).

Then, qRT-PCR was performed for testing the expression profiles of the candidate *PhTALE* genes among the different parts of a single bamboo internode. As a result, all the *PhBLH* genes (except *PhBLH16*) had high expression levels in the upper part of the internode (Figure 8). Similar expression profiles appeared in *PhKNOX* genes (Figure 9) as well. Compared with middle and basal parts, all the candidate *PhTALE* genes (except *PhBLH16*) had obviously higher expression levels in the upper part. The above results suggested that these candidate *PhTALE* genes may positively regulate the lignification of the internode during the fast-growth stage of moso bamboo. Then, cell-wall-related genes that appeared in both brown and blue modules were selected for qRT-PCR analysis (Figure 10A). Interestingly, 8 genes were annotated as IRX-like genes among the 22 selected genes, and all had significantly higher expression levels in the upper part (Figure 10E). The IRX-like genes are involved in xylan backbone extension. In addition, all pectin and cellulose biosynthesis genes were found to have obviously higher expression levels in the upper part of the internode (Figure 10B,D). One lignin-related gene, *PhNAC43-4*, also showed a higher expression level in the upper part (Figure 10C). All above results suggested that the upper part of the internode had higher lignification and the candidate *PhTALEs* may participate in regulating secondary cell wall deposition.

### 2.7. Monosaccharide Composition in Different Parts of the Internode

Xylan is the main hemicellulose of plant secondary cell walls. Here, to clarify the synthesis of xylan in the three parts of the single internode, we quantified the monosaccharide compositions in the three parts. The contents of arabinose, fucose, galactose, glucose, mannose, xylose, glucuronic acid, rhamnose, and galacturonic acid were measured (Figure 11). Among these monosaccharides, arabinose, glucose, mannose, galactose, and galacturonic acid had higher contents in the basal part of the bamboo internode. Rhamnose was not detected in the internode. Only xylose and glucuronic acid had obviously higher contents in the upper part, and the glucuronic acid was only detected in the upper part. The xylose content in the upper part was 3.5 and 4.1 times of that in the middle part and basal part, respectively. These results suggested that more xylan synthesis takes place in the upper part of the bamboo internode.

## 3. Discussion

Bamboos belong to Poaceae, one of the fastest growing plants on earth. Bamboo plants can be classified into herbaceous bamboos and woody bamboos. Woody bamboos are thought to have split from herbaceous bamboos. Furthermore, the evolutionary process of bamboos is thought to be a reticulate evolution [22]. Compared with herbaceous bamboos, woody bamboos are higher in both height and lignification. By comparing the genomes of herbaceous and woody bamboos, a remarkable increase of cellulose- and lignin-related genes was found in woody bamboos and were thought to be relevant to the higher lignification in woody bamboos [22]. The homeobox gene family is a large transcription factor gene family and found distributing in almost all eukaryotes. The proliferation and diversities of homeobox genes are thought to be consistent with the evolution of organism structural and developmental complexity [2]. Here, homeobox genes in *P. edulis*, *B. amplexicaulis*, and *O. latifolia* were identified and compared. Obvious expansion was found in the TALE subclass of woody bamboo compared with herbaceous bamboo. When compared with the TALE subfamily of rice and maize, similar expansion appeared in woody bamboos.

Gene duplication is a driver of plant evolution in morphogenesis. In plants, gene duplication can be mainly divided into whole-genome duplication (WGD) and single-gene duplication. WGD is thought to coincide with speciation [27,28]. A large number of paralogs will be generated after WGD, while only part will be retained during the evolution [29]. By investigating Arabidopsis’s genome, it was found that paralogs participating in signaling and transcriptional regulation are more often retained. The quantity of genes related to transcriptional regulation is positively correlated with plant morphological complexity. Transcription-associated proteins are key factors affecting plant morphogenesis [30,31]. Compared with herbaceous bamboos, great changes happened in woody bamboos. Woody bamboos have higher culms and increased lignification. Through comparative genomics research between herbaceous and woody bamboos, the absolute copy number of genes related to cell wall lignification, such as cellulose- and lignin-related genes, was found to be remarkably increased in woody bamboos. The increase of related genes was also found relevant to the origin of woody traits in woody bamboos [22]. These research reports indicated that retaining of a large number of paralogs of cell-wall-related genes after WGD or polyploidization may be the method of evolution in woody bamboos [22]. Moso bamboo, a classic woody bamboo, has the maximum area in China. Furthermore, the high-quality genomic map of moso bamboo has been released. To further investigate the TALE family, moso bamboo was adopted as the research target here. In moso bamboo, WGD is the main force for the expansion of the TALE gene family. Compared with *O. latifolia* and rice, more paralogs of TALE genes were retained in moso bamboo after WGD.

The TALE gene family is the oldest subclass in the homeobox gene family and has been widely studied in Arabidopsis and rice. The TALE family can be then classified into BLH and KNOX subfamilies. KNOX proteins always function by forming heterodimers with BLH proteins [4,5]. Moso bamboo and rice are members of Poaceae and were found to have high genomic synteny [32]. Furthermore, the cell wall components of grass species and many dicots are different. Grass species have more xylan in the cell walls [33]. Thus, rice was thought to be a very useful model plant in moso bamboo research. To predicate the function of *PhTALE* genes, the collinearity of the homeobox gene family between moso bamboo and rice was performed. High synteny was found between the homeobox gene family of moso bamboo and rice. Furthermore, some *OsTALE* genes had more than two orthologous genes in moso bamboo genome. *OsTALE* genes, such as *SH5, OsKNAT7*, and *OSH15*, playing central roles in regulating secondary cell wall deposition were found to have several orthologous genes in moso bamboo. Among these genes, *OSH15* have four orthologs, and the other two *OsTALE* genes (*SH5* and *OsKNAT7*) have two orthologs. By constructing co-expression networks, a total of 29 *PhTALE* genes were clustered into the modes that have strong correlation with plant cell wall development. The orthologs of *SH5 (PhBLH11), OsKNAT7 (PhKNOX7* and *PhKNOX9)*, and *OSH15 (PhKNOX2*, *PhKNOX3*, and *PhKNOX5)* in moso bamboo were found to be clustered in the mode that correlated with secondary cell wall development. These 29 genes also had high expression levels in tissues of bamboo culms or shoots with higher lignification. These results suggested that *TALE* genes in moso bamboo may be conserved in regulating secondary cell wall development.

Moso bamboo is known as one of the fastest growing plants on earth. During the fast-growing stage, the simultaneous elongation growth of several internodes upbuild the fast elongating of the whole culm. Recently, more and more attention were paid on the development of a single internode in bamboos. The single internode of *Bambusa multiplex* (*Lour*.) *Raeusch*. *ex Schult* has been studied in detail [34]. In this study, the cell division zone (DZ) and the cell elongation zone (EZ) were identified. In rice, the internode can also be divided into DZ, EZ, and maturation zone (MZ). From the basal to the upper part, the developmental stages of the rice internode show a gradient distribution, and secondary cell wall deposits are most abundant in the upper part (MZ) [35]. Herein, similar gradient distribution of developmental stages was also found in the single internode of moso bamboo at the fast-growing stage. Through toluidine blue staining, thicker cell walls were found in the upper part of the internodes. The contents of cellulose and lignin also supported the gradient distribution of developmental stages in the single internode of moso bamboo. Through qRT-PCR, all candidate *PhTALE* (except *PhBLH16*) genes were found having significantly higher expression levels in the upper part of the internode. These results suggested that the deposition of the secondary cell wall is more active in the up-part, and these candidate *PhTALE* (except *PhBLH16*) genes may be positive regulators of the deposition during the fast growth of moso bamboo internode.

Some reported *OsTALE* genes, such as *OsKNAT7, OSH15*, and *SH5*, were reported to inhibit lignin biosynthesis and play negative roles in the secondary cell wall deposition in rice [18,19]. In Arabidopsis, *AtKNAT1* and *AtKNAT3* are positive regulators and *AtKNAT7* and *AtBLH6* are negative regulators in secondary cell wall deposition [11,17]. In cotton, over-expression *GhKNAT7* and *GhBLH6* was found to significantly inhibit the lignin synthesis in Arabidopsis [16]. To further analyze the function of candidate *PhRALE* genes, we identified and annotated the cell-wall-related genes that co-expressing with the candidate *PhTALE* genes. A total of 22 genes were found appearing in both modes (brown and blue). Among the 22 genes, 8 genes were annotated as IRREGULAR XYLEM(IRX)-like genes. According to previous reports, IRX9, IRX10, IRX9L, IRX10, IRX10L, IRX14, and IRX14L are key enzymes in xylan biosynthesis [14]. The expression profiles of these *PhIRX*-like genes are similar to that of the candidate *PhRALE* genes. All eight *PhIRX*-like genes had high expression levels in the upper part. Detection of monosaccharide composition also indicated that xylose content in the upper part was significantly higher than that in middle and basal parts. These results suggested that more xylan was biosynthesized in the upper part of the internode in moso bamboo. Xylan is the main hemicellulose of secondary cell walls in grasses, and deposition of xylan increases the recalcitrance of cell wall [33]. In Arabidopsis, *AtKNAT7* was found to promote the biosynthesis of xylan by interacting with *IRX9* and *IRX14* [15]. In poplar, over-expression of *ANK1* (a class Ⅰ KNOX homeobox gene) can also promote the xylan synthesis in the stem [36]. However, *OsKNAT7* inhibits the xylose synthesis in rice [19]. In grasses, xylan structure is known as glucuronoarabinoxylan, which is different from that in dicot plants [33]. Here, glucuronic acid was detected only in the upper part of the internode. These results suggested that more xylan deposited in the upper part of the internode at this stage, and candidate *PhTALE* (*PhBLH11*, *PhKNOX11*, *PhKNOX13*, *PhKNOX2*, *PhKNOX3*, *PhKNOX5*, *PhKNOX7*, and *PhKNOX9*) genes may be involved in regulating the xylan synthesis in the secondary cell wall during the fast-growing stage.

## 4. Materials and Methods

### 4.1. Identification and Classification of Homeobox Genes in Bamboos

Genes with the homeobox domain are known as homeobox genes. To identify the homeobox genes in *P. edulis*, *B. amplexicaulis*, and *O. latifolia*. HMMER 3.0 software was adopted (Version 3.0; Robert D Finn, USA, 2015) [37]. Hidden Markov Models (HMM) of homeobox domain (PF00046) and ZF-HD dimerization domain (PF04770) were downloaded from the Pfam database (http://pfam.xfam.org/; accessed on 29 September 2020). After searching by HMMER with HMMs, the protein sequences of all candidate homeobox genes were uploaded to Pfam and SMART (http://smart.embl-heidelberg.de/; accessed on 26 October 2020) database for further validation to determine the presence of homeobox domain. Genes without homeobox domain or ZF-HD dimerization domain were deleted [38]. The genome data of *P. edulis* was downloaded from BambooGDB (http://bamboo.bamboogdb.org/; accessed on 3 July 2019), *O. latifolia* and *B. amplexicaulis* were download from another bamboo genome database (http://www.genobank.org/bamboo; accessed on 4 December 2019).

Homeobox family genes have 14 subclasses and are distinguished by homeobox domain and characteristic codomains. Herein, we classified homeobox gene families of the three bamboo plants through constructing phylogenetic trees with the full-length amino acid sequences of homeobox genes. Protein structures of these homeobox genes were also taken into account. Before constructing the phylogenetic trees, ProtTest (Version 3.0; University of Vigo, Vigo, Spain, 2011) was used for computing the best amino acid substitution model with full-length protein sequences [39]. Then, RaxML-ng (Heidelberg Institute for Theoretical Studies, Heidelberg, Germany, 2019) was adopted for constructing phylogenetic trees with maximum-likelihood (ML) method under the best models selecting by ProtTest [40]. Protein sequences of rice homeobox gene family was used as a reference when constructing phylogenetic trees. The bootstrap replication was set as 1000. Evolview v3 (version 3.0; Huazhong University, Wuhan, China, 2019) was used for visualization and management of phylogenetic trees.

### 4.2. Collinearity Analysis and Duplication Mode Identification

To identify different duplication models of gene duplication in moso bamboo, multiple collinearity scan toolkit (MCScanX; University of Georgia, Athens, GA, USA, 2012) was adopted [41]. First, an all-to-all BLASTP was performed among all protein sequences from the moso bamboo genome (E < 1 × 10^−5^, top 5 matches). Then, the BLASTP results and annotation gff3 files were used as input files of MCScanX to determine duplication modes in moso bamboo homeobox gene family. Different duplication modes were defined as previously described by Wang et al. The collinearity between moso bamboo and rice was also analyzed through MCScanX. Circos was used for visualization (Canada′s Michael Smith Genome Sciences Center, Vancouver, BC, Canada, 2009) [42].

### 4.3. Calculation of Ka/Ks Ratios

The valid gene pairs coming from different duplication modes were used to calculate the non-synonymous (*Ka*) and synonymous (*Ks*) substitution rates. Pairwise alignment of protein sequences for each duplicate gene pair were performed by MAFFT (Version 7; Osaka University, Osaka, Japan, 2013). Then, *KaKs*_Calculator 2.0 (Version 2; Chinese Academy of Sciences, Beijing, China, 2010) was used to calculate the *Ka*, *Ks*, and *Ka*/*Ks* ratios with gamma-MYN method [43].

### 4.4. Construction of Weight Gene Co-Expression Networks

The transcriptome sequencing data used for constructing weight gene co-expression networks (WGCNA, University of California, Los Angeles, CA, USA, 2008) were downloaded from NCBI Sequence Read Archive (SRA) (https://www.ncbi.nlm.nih.gov/sra/; accessed on 23 July 2020) [44]. A total of 29 data samples from 2 projects were adopted in this study (Appendix A). The information of these data is provided in Appendix A. Data from the 2 projects were used to construct networks, respectively. “Transcripts Per Kilobase of exon model per Million mapped reads” (TPM) was adopted for quantifying the expression level of each gene. The TPM counts of these data were used as the input file of WGCNA. Here, step-by-step network construction function was adopted for constructing the network. Genes with too many missing values were deleted; 27867 and 28835 genes were finally used for constructing co-expression networks. Power value, minModuleSize, and deepSplit were set as 7, 30, and 2, respectively. Visualization software used here was cytoscape_v.3.8.2 (Version 3.8.2; Institute for Systems Biology, Seattle, WA, USA, 2003). ClusterProfile (Version 4.0; Southern Medical University, Guangzhou, China) was used for gene ontology (GO) enrichment analysis with *p* < 0.05 [45].

### 4.5. Plant Material and Anatomical Structure Analysis

Moso bamboo shoots with height of ~3.5 m were selected as the research object. Then, the upper, middle, and basal parts of the 14th internode with length of 15–17 cm were collected, respectively. Every part had at least 3 biological replicates. Samples for isolating total RNA were stored in liquid nitrogen.

To examine the cell wall developmental stage, three parts of the selected internode were sampled for anatomical structural analysis. Here, toluidine blue was used to stain sections of the three internode parts following the approach described by Turner. Lignified cell walls will be stained blue.

### 4.6. Measurement of Lignin, Cellulose, and Monosaccharide Composition

To observe the situation of cell wall at the upper, middle, and basal part of the internodes, we measured the lignin and cellulose content of the three parts. Every part had at least 3 biological replicates. Samples of the three parts were ground with liquid nitrogen. Then, the cellulose content was quantified by anthrone-sulphuric acid method as previously described [46]. Lignin content was measured colorimetrically quantified by acetyl bromide method as previously described [47].

Monosaccharide composition in different parts of the single internode was measured following the method reported by Hu et al. [48]. Briefly, 2 mg powder sample was hydrolyzed with 1 mL trifluoroacetic acid (TFA, 4 M) at 120 °C and then blow dried with nitrogen. The hydrolysates were then derivatized with 0.5 M 1-phenyl-3-methyl-5-pyrazolone (PMP) and 0.3 M NaOH at 70 °C 60 min. The HPLC system (Waters) equipped with a SHISEIDO C18 (SHISEIDO, Tokyo, Japan) column was used to separate the different monosaccharides. Monosaccharide standards were included as references in the experiment.

### 4.7. Total RNA Isolation and qRT-PCR Analysis of the Selected Genes

Total RNA of the internodes was isolated by an RNA extraction kit (Tiangen, Beijing, China). Every part had at least 3 biological replicates. The reverse transcription of the RNA was performed by a PrimerScript RT reagent kit (TaKaRa, Dalian, China). The cDNA used for quantitative real-time PCR (qRT-PCR) analysis was diluted 18-fold. Primers of the selected genes used for qRT-PCR were designed with Primer 5.0 and are displayed in Appendix A. The cycling conditions were set as follows: 95 °C for 30 s, 40 cycles at 95 °C for 10 s, 60 °C for 30 s, and 1 cycle at 95 °C for 10 s, 60 °C for 60 s, 95 °C for 15 s to create a melting curve. Nucleotide tract-binding protein gene (NTB) was chosen to be an internal control [49]. The 2^−^^△△Ct^ method was used for computing the relative expression levels of the candidate genes [50].

## Figures and Tables

**Figure 1 ijms-23-04112-f001:**
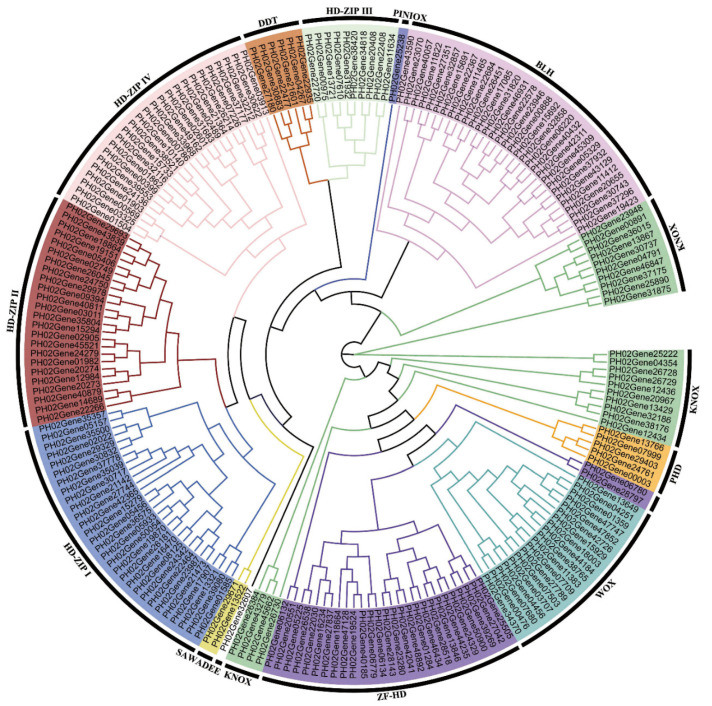
The phylogenetic tree of moso bamboo homeobox genes. The phylogenetic tree was constructed via RaxML-ng on the basis of full-length amino acid sequences of homeobox genes with maximum-likelihood (ML) method under the best models selecting by ProtTest. Bootstrap analysis was conducted with 1000 replicates.

**Figure 2 ijms-23-04112-f002:**
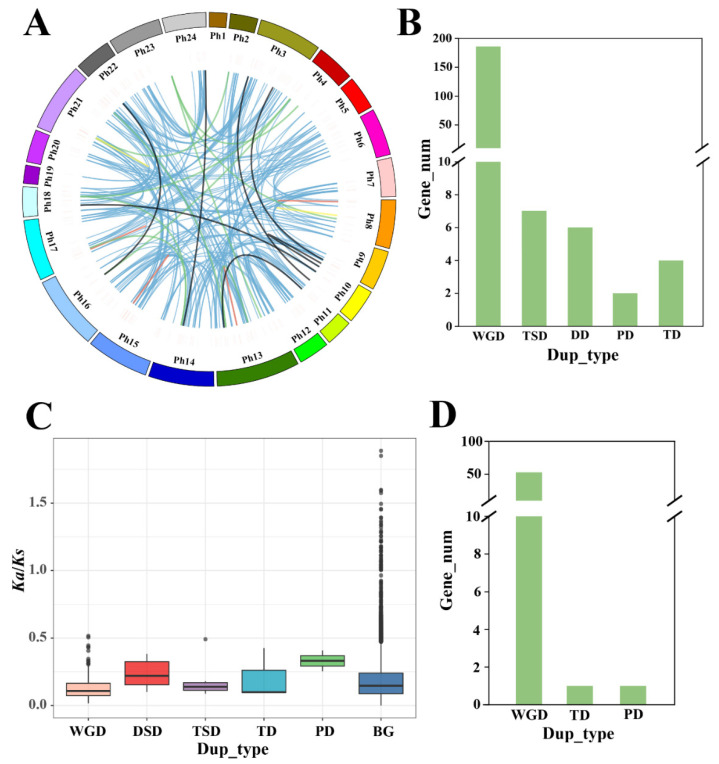
The expansion and evolution analysis of homeobox genes in moso bamboo. (**A**) The distribution of duplicate gene pairs in moso bamboo 24 chromosomes. The gene pairs originating from WGD are linked by blue lines, proximal pairs by orange lines, dispersed pairs by green, tandem pairs by red, and DNA-transposed pairs by black. (**B**) The number of duplicate gene pairs of different modes in homeobox gene family. (**C**) Values of *Ka/Ks* for gene pairs of different duplication modes and background genes. WGD: whole-genome duplication; PD: proximal duplication; TD: tandem duplication; TSD: DNA-transposed duplication; DD: dispersed duplication; BG: background genes. (**D**) The number of duplicate gene pairs of different modes in TALE gene family.

**Figure 3 ijms-23-04112-f003:**
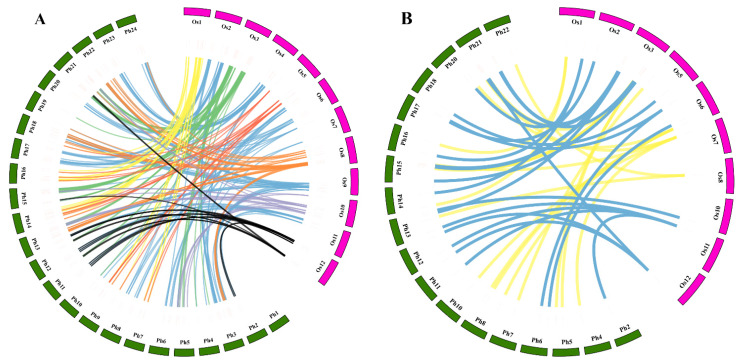
Synteny analysis of homeobox genes in moso bamboo and rice. (**A**) Synteny analysis of homeobox genes between moso bamboo and rice. The chronosomes of rice were marked with different colors. (**B**) Synteny analysis of TALE genes between moso bamboo and rice. KNOX genes were marked in yellow; BLH genes were marked in blue.

**Figure 4 ijms-23-04112-f004:**
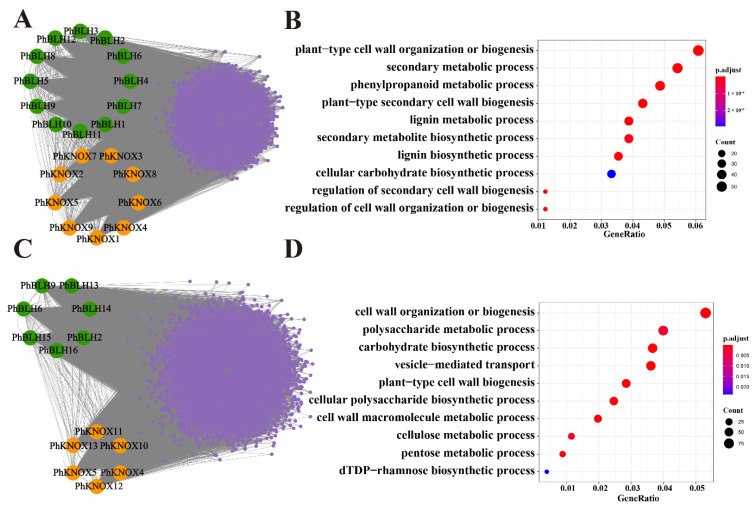
Gene co-expression analysis of *PhTALE* genes in moso bamboo. (**A**) *PhTALE* genes involved in the brown module. *PhBLH*s are marked with green color, *PhKNOX*s are marked with orange color. Genes co-expressing with *PhTALE*s are marked with purple. (**B**) Functional enrichment map of gene ontology (GO) terms for genes in the brown module. (**C**) *PhTALE* genes involved in the blue module. *PhBLH*s are marked with green color, *PhKNOX*s are marked with orange color. Genes co-expressing with *PhTALE*s are marked with purple. (**D**) Functional enrichment map of gene ontology (GO) terms for genes in the blue module.

**Figure 5 ijms-23-04112-f005:**
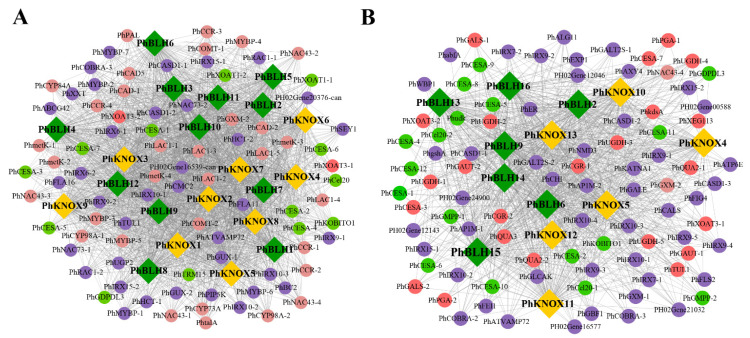
Cell-wall-related genes that co-express with *PhTALE* genes in moso bamboo. (**A**) Cell-wall-related genes that co-express with *PhTALE* genes in the brown module. (**B**) Cell-wall-related genes that co-express with *PhTALE* genes in the blue module. Cellulose-related genes are marked with light green, pectin-related genes are marked with light red, lignin-related genes are marked with light pink, other cell-wall-related genes are marked with purple.

**Figure 6 ijms-23-04112-f006:**
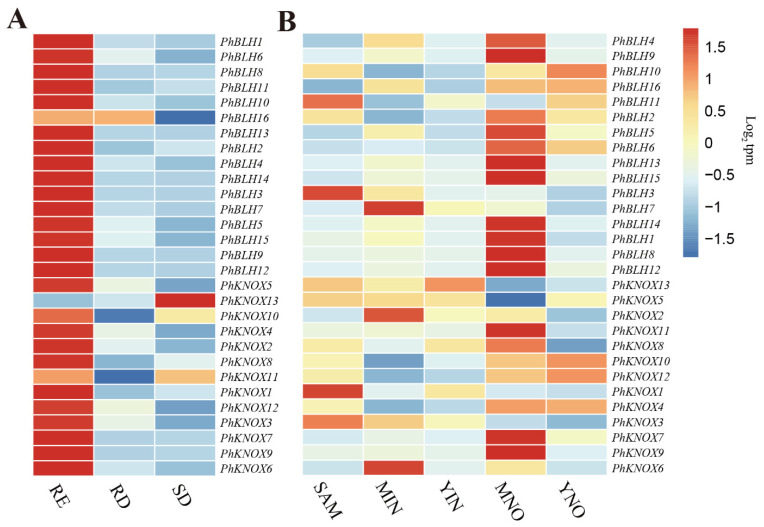
Expression pattern analysis of candidate *PhTALE* genes. (**A**) The expression profile of candidate *PhTALE* genes in different growth stage of moso bamboo. SD, starting of cell division; RD, rapid division; RE, rapid elongation. (**B**) The expression profile of candidate *PhTALE* genes in different parts of young moso bamboo shoot. Data are shown as log_2_ transcripts per million (log_2_ tpm). (**A**,**B**) uses the same color legend. SAM, shoot apical meristem; YIN, young internode; YNO, young node; MIN, basal mature internode; MNO, basal mature node.

**Figure 7 ijms-23-04112-f007:**
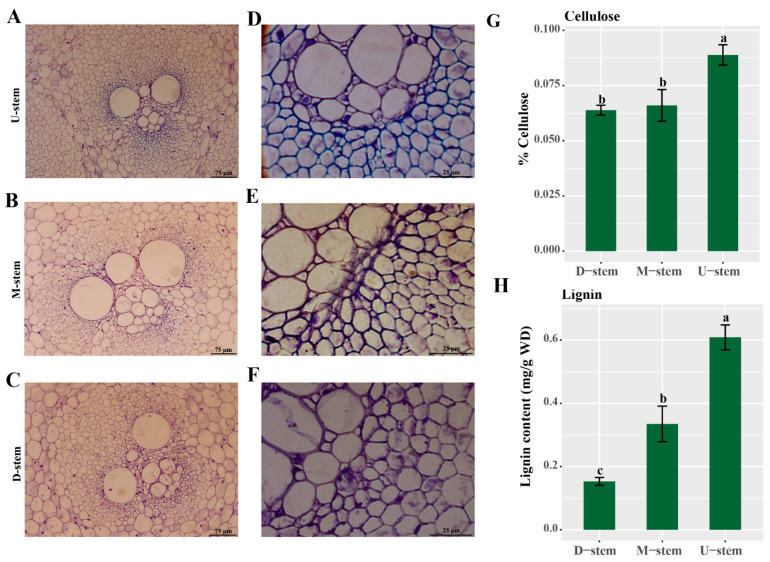
The lignification in the upper part of the moso bamboo single internode is higher. (**A**,**D**) Cross sections of the upper part in moso bamboo single internode. (**B**,**E**) Cross sections of the middle part of moso bamboo single internode. (**C**,**F**) Cross sections of the basal part of moso bamboo single internode. (**G**) Cellulose content in different parts of the moso bamboo internode. (**H**) Lignin content in different parts of the moso bamboo internode.Scale bars represent 75 μm (A–C) and 25 μm (D–F). Error bars represent standard deviation (SD). One-way ANOVA test was used to identify the differences, and columns with the same letter are not significantly different (*p* < 0.05). D-stem, basal part of the internode; M-stem, middle part of the internode; U-stem, upper part of the internode.

**Figure 8 ijms-23-04112-f008:**
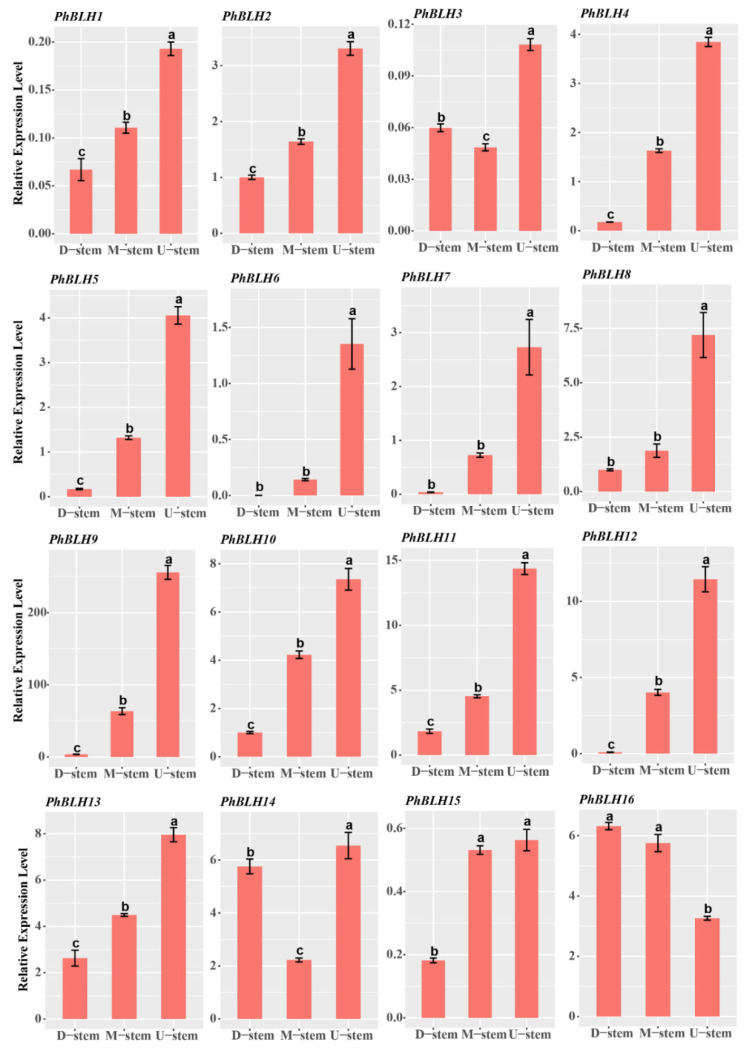
Expression profiles of candidate *PhBLH* genes in different parts of moso bamboo internode. Error bars represent standard deviation (SD). One-way ANOVA test was used to identify the differences, and columns with the same letter are not significantly different (*p* < 0.05). D-stem, basal part of the internode; M-stem, middle part of the internode; U-stem, upper part of the internode.

**Figure 9 ijms-23-04112-f009:**
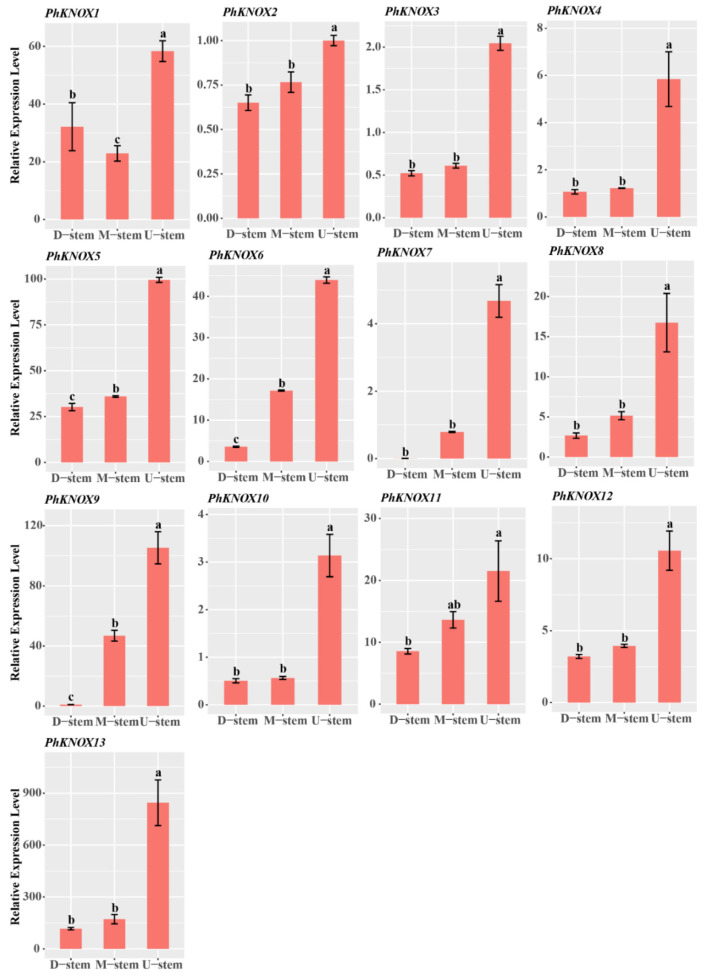
Expression profiles of candidate *PhKNOX* genes in different parts of moso bamboo internode. Error bars represent standard deviation (SD). One-way ANOVA test was used to identify the differences, and columns with the same letter are not significantly different (*p* < 0.05). D-stem, basal part of the internode; M-stem, middle part of the internode; U-stem, upper part of the internode.

**Figure 10 ijms-23-04112-f010:**
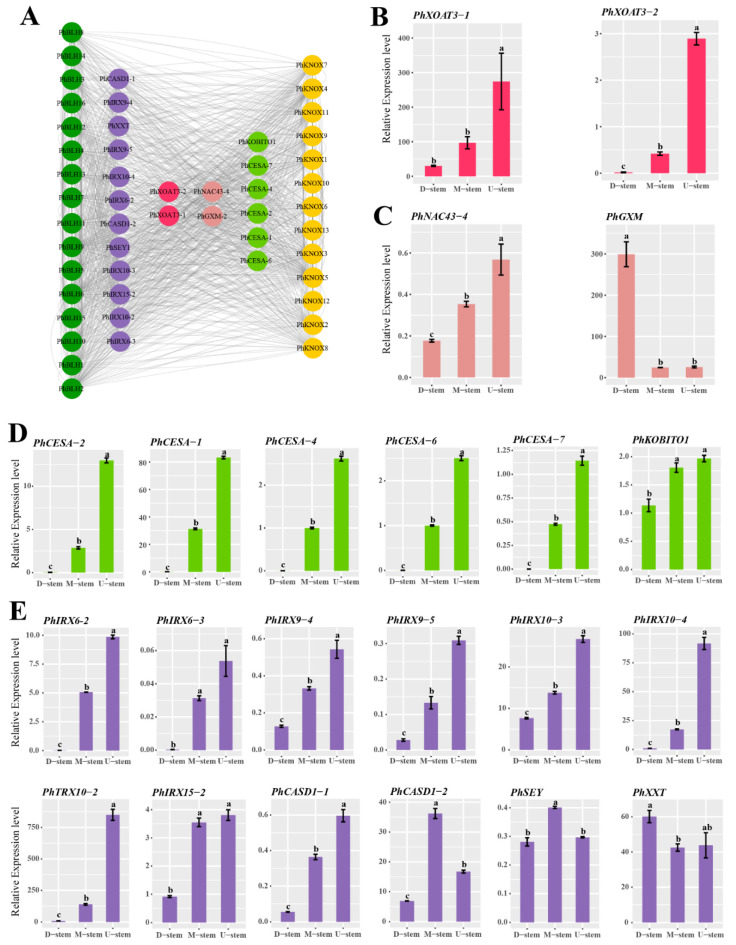
Expression profiles of cell-wall-related genes common in brown and blue modules. (**A**) Cell-wall-related genes common in brown and blue modules. (**B**) Expression profiles of genes related to pectin in different parts of moso bamboo internode. (**C**) Expression profiles of genes related to lignin in different parts of moso bamboo internode. (**D**) Expression profiles of genes related to pectin in different parts of moso bamboo internode. (**E**) Expression profiles of other genes that related to cell wall in different parts of moso bamboo internode. Error bars represent standard deviation (SD). One-way ANOVA test was used to identify the differences, and columns with the same letter are not significantly different (*p* < 0.05). D-stem, basal part of the internode; M-stem, middle part of the internode; U-stem, upper part of the internode. Cellulose-related genes are marked with light green, pectin-related genes are marked with light red, lignin-related genes are marked with light pink, other cell-wall-related genes are marked with purple.

**Figure 11 ijms-23-04112-f011:**
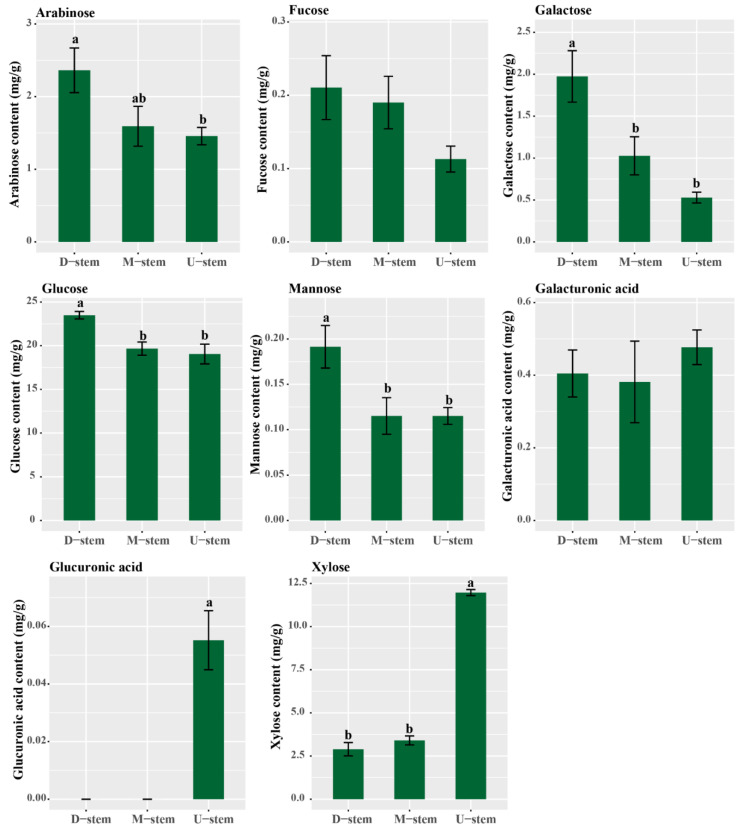
Monosaccharide composition in different parts of the moso bamboo internode. Error bars represent standard deviation (SD). One-way ANOVA test was used to identify the differences, and columns with the same letter are not significantly different (*p* < 0.05). D-stem, basal part of the internode; M-stem, middle part of the internode; U-stem, upper part of the internode.

**Table 1 ijms-23-04112-t001:** Classification of homeobox proteins retrieved from different plants.

Class	Eudicots			Monocots					Bryophyta Moss	Lycopodiophyta Sm
	At	Poplar	Soybean	Rice	Maize	Moso	Oly	Bam		
HD-ZIPI	17	22	35	14	24	29	13	34	17	4
HD-ZIPII	10	16	27	14	17	24	14	31	7	2
HD-ZIPIII	5	8	12	7	8	10	6	11	5	3
HD-ZIPIV	16	15	31	12	21	25	11	25	4	4
PLINC	14	17	51	11	17	29	17	41	11	5
WOX	16	19	33	15	16	19	12	24	3	6
KNOX	8	15	28	12	13	24	11	32	5	5
BLH	13	19	34	14	17	31	15	34	4	2
DDT	4	7	13	3	4	6	2	6	3	2
PHD	2	4	6	2	5	5	2	3	2	1
NDX	1	2	1	1	1				2	1
LD	1	2	1	1	1			1	1	1
PINTOX	1	1	2	1	1	1	1	1	1	1
SAWADEE	2	1	2	3	3	2	1	3	1	2
Total	110	148	276	110	148	205	105	246	66	45

## Data Availability

In this section, transcriptional data were downloaded from NCBI, and physiological and anatomic metabolic data were measured by the authors themselves.

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
