# Peer review of "Genome-Wide Identification, Expansion, and Evolution Analysis of Homeobox Gene Family Reveals TALE Genes Important for Secondary Cell Wall Biosynthesis in Moso Bamboo (Phyllostachys edulis)"

_ijms, 2022, doi:10.3390/ijms23084112_

Round 1

Reviewer 1 Report

Moso Bamboo is one of the fastest-growing plants on earth, and an important sustainable timber resource and ecological conservation resource. The reasons why bamboo grows so fast are very interesting and valuable. 

The authors analyzed TALE gene family of Moso bamboo(Phyllostachys edulis), Olyra latifolia and Bonia amplexicaulis. They studied the duplication situation, evolution pressures, co-expression relationships, GO annotation, and RT-PCR validation. Overall, the methods seem to be fine, the results make sense.

Here are some comments:

  1. The introduction may include some previous research results of the Bamboo whole-genome duplication situation, cause they have results based on the duplicated genes.  We need to know the WGD history of bamboo to tell whether the homeobox genes are different or not. I've seen some lines in the discussion section, but I still didn't know the average duplication level of bamboo genes through the words.
  2. Figure 2C. Ka/Ks analysis. I suggest you can add a column of Ka/Ks values for background genes. Also, you could try to calculate the divergence time of these three bamboos or the duplicated genes through Ks values. 
  3. Figure 4BD. many redundant GO terms could be merged on some standard, the simplify function of ClusterProfiler or manual filtering could be considered. 
  4. Figure 4AC, The color of the gene type needs to be indicated in the legend, and I strongly recommend that you avoid using blue and brown in the networks that you already used as WGCNA module names which may confuse readers. Same problems as Figure 5 and Figure 10A.
  5. Figure 6, scaled TPM color needs an explanation in legend. Are A and B using the same color legend?
  6. Figure 7,8,9,10. What kind of test you had used. What specific n of each sample?  

Thanks.

Author Response

The authors analyzed TALE gene family of Moso bamboo (Phyllostachys edulis), Olyra latifolia and Bonia amplexicaulis. They studied the duplication situation, evolution pressures, co-expression relationships, GO annotation, and RT-PCR validation. Overall, the methods seem to be fine, the results make sense.

Major Compulsory Revisions

  1. The introduction may include some previous research results of the Bamboo whole-genome duplication situation, because they have results based on the duplicated genes. We need to know the WGD history of bamboo to tell whether the homeobox genes are different or not. I've seen some lines in the discussion section, but I still didn't know the average duplication level of bamboo genes through the words.

Response:

--We thank the reviewer for this comment.

--As suggested by the reviewer, we had added the explanation about the WGD history of moso bamboo in the introduction part and result part “Expansion and evolution events of homeobox gene family in moso bamboo”.

Among these bamboos, the whole-genome sequencing of moso bamboo (Phyllostachys edulis) had been released. An ancient whole-genome duplication (WGD) event that occurred at 7-12 million years ago was thought important in the evolution of moso bamboo [22, 23]. (Lines 92-95)

The homeobox gene family's Ka/Ks value distribution is similar to that of the moso bamboo whole genome. (Lines 167-168)

  1. Figure 2C. Ka/Ks analysis. I suggest you can add a column of Ka/Ks values for background genes. Also, you could try to calculate the divergence time of these three bamboos or the duplicated genes through Ks values.

Response:

--We thank the reviewer for this comment.

-- As suggested by the reviewer, we had added a column of Ka/Ks values for the WGD gene pairs of the moso bamboo whole-genome (Indicated in BG). In addition, the whole genome sequencing of Olyra latifolia and Bonia amplexicaulis are the draft genome. The chromosome location information of genes is unavailable, so we can’t calculate the divergence time of these three bamboos or the duplicated genes through Ks values.

The homeobox gene family's Ka/Ks value distribution is similar to that of the moso bamboo whole genome. (Lines 167-168)

  1. Figure 4BD. many redundant GO terms could be merged on some standard, the simplify function of ClusterProfiler or manual filtering could be considered.

Response:

--We thank the reviewer for this comment.

-- As suggested by the reviewer, we had removed the redundant GO terms by using the simplify function of ClusterProfiler and redrawn the GO enrichment analysis result in Figure 4B and D.  

Furthermore, genes in brown module were found enriching in biology processes related to secondary cell wall development, such as lignin biosynthetic process, lignin meta-bolic process, plant-type secondary cell wall biogenesis, and regulation of secondary cell wall biogenesis (Figure 4 B). While in blue module, genes were found enriching in cell wall organization or biogenesis, carbohydrate and polysaccharide biosynthetic processes (Figure 4D). (Lines 226-231)

  1. Figure 4AC, The color of the gene type needs to be indicated in the legend, and I strongly recommend that you avoid using blue and brown in the networks that you already used as WGCNA module names which may confuse readers. Same problems as Figure 5 and Figure 10A.

Response:

--We thank the reviewer for this comment.

-- As suggested by the reviewer, we had added the indication of the color of the gene type in the legend. The colors, blue and brown, were also replaced with other colors in Figure 4 A and C, Figure 5 and Figure 10A.

Figure 4. Gene co-expression analysis of PhTALE genes in moso bamboo. (A) PhTALE genes in-volved in the brown module. PhBLHs had been marked with green color, PhKNOXs were marked with orange color. Genes co-expressing with PhTALEs had been marked with purple. (B) Functional enrichment map of gene ontology (GO) terms for genes in the brown module. (C) PhTALE genes involved in the blue module. PhBLHs had been marked with green color, PhKNOXs were marked with orange color. Genes co-expressing with PhTALEs had been marked with purple. (D) Functional enrichment map of gene ontology (GO) terms for genes in the blue module. (Lines 249-255)

Figure 5. Cell wall related genes that co-expression with PhTALE genes in moso bamboo. (A) Cell wall related genes that co-expression with PhTALE genes in the brown module. (B) Cell wall related genes that co-expression with PhTALE genes in the blue module. Cellulose related genes had been marked with lightgreen, Pectin related genes had been marked with lightred, Lignin related genes had been marked with lightpink, Other cell wall related genes had been marked with purple. (Lines 260-264)

Figure 10. Expression profiles of cell wall related genes common in brown and blue modules. (A) Cell wall related genes common in brown and blue modules. (B) Expression profiles of genes related to pectin in different parts of moso bamboo internode. (C) Expression profiles of genes related to lignin in different parts of moso bamboo internode. (D) Expression profiles of genes related to pectin in different parts of moso bamboo internode. (E) Expression profiles of other genes that related to cell wall in different parts of moso bamboo internode. Error bars represent standard deviation (SD). One-way Anova test was used was used to identify the differences, and columns with the same letter are not significantly different (P < 0.05). D-stem, down-part of the internode; M-stem, Mid-dle-part of the internode; U-stem, up-part of the internode. Cellulose related genes had been marked with lightgreen, Pectin related genes had been marked with lightred, Lignin related genes had been marked with lightpink, Other cell wall related genes had been marked with purple. (Lines 356-366)

  1. Figure 6, scaled TPM color needs an explanation in legend. Are A and B using the same color legend?

Response:

--We thank the reviewer for this comment.

-- As suggested by the reviewer, we had added the explanation of TPM in the legend. Figure 6A and B using the same color legend.

Figure 6. Expression patterns analysis of candidate PhTALE genes. (A) The expression profile of candidate PhTALE genes in different growth stage of moso bamboo. SD, starting of cell division; RD, rapid division; RE, rapid elongation. (B) The expression profile of candidate PhTALE genes in different parts of young moso bamboo shoot. Data are shown as log2 (transcripts per million) (log2 tpm). A and B using the same color legend. SAM, shoot apical meristem; YIN, young internode; YNO, young node; MIN, basal mature internode; MNO, basal mature node. (Lines 293-298)

  1. Figure 7,8,9,10. What kind of test you had used. What specific n of each sample?

Response:

--We thank the reviewer for this comment.

-- One-way Anova test was used to identify the differences in this study, and we had added relative description in the legends of Figure 7,8,9 and 10. Each sample had at least 3 biological replicates.

Figure 7. The lignification in the up-part of the moso bamboo single internode is higher. (A and D) Cross sections of the up-part in moso bamboo single internode. (B and E) Cross sections of the middle-part in moso bamboo single internode. (C and F) Cross sections of the down-part in moso bamboo single internode. (G) Cellulose content in different parts of the moso bamboo internode. (H) Lignin content in different parts of the moso bamboo internode. Error bars represent standard de-viation (SD). One-way Anova test was used to identify the differences, and columns with the same letter are not significantly different (P < 0.05). D-stem, down-part of the internode; M-stem, Middle-part of the internode; U-stem, up-part of the internode. (Lines 333-341)

Figure 8. Expression profiles of candidate PhBLH genes in different parts of moso bamboo internode. Error bars represent standard deviation (SD). One-way Anova test was used to identify the dif-ferences, and columns with the same letter are not significantly different (P < 0.05). D-stem, down-part of the internode; M-stem, Middle-part of the internode; U-stem, up-part of the internode. (Lines 343-347)

Figure 9. Expression profiles of candidate PhKNOX genes in different parts of moso bamboo in-ternode. Error bars represent standard deviation (SD). One-way Anova test was used to identify the differences, and columns with the same letter are not significantly different (P < 0.05). D-stem, down-part of the internode; M-stem, Middle-part of the internode; U-stem, up-part of the internode. (Lines 350-354)

Figure 10. Expression profiles of cell wall related genes common in brown and blue modules. (A) Cell wall related genes common in brown and blue modules. (B) Expression profiles of genes related to pectin in different parts of moso bamboo internode. (C) Expression profiles of genes related to lignin in different parts of moso bamboo internode. (D) Expression profiles of genes related to pectin in different parts of moso bamboo internode. (E) Expression profiles of other genes that related to cell wall in different parts of moso bamboo internode. Error bars represent standard deviation (SD). One-way Anova test was used to identify the differences, and columns with the same letter are not significantly different (P < 0.05). D-stem, down-part of the internode; M-stem, Middle-part of the internode; U-stem, up-part of the internode. Cellulose related genes had been marked with light-green, Pectin related genes had been marked with lightred, Lignin related genes had been marked with lightpink, Other cell wall related genes had been marked with purple. (Lines 357-367)

Figure 11. Monosaccharide composition in different parts of the moso bamboo internode. Error bars represent standard deviation (SD). One-way Anova test was used to identify the differences, and columns with the same letter are not significantly different (P < 0.05). D-stem, down-part of the internode; M-stem, Middle-part of the internode; U-stem, up-part of the internode. (Lines 381-385)

Moso bamboo shoots with height of ~ 3.5 m were select as the research object. Then, the upper, middle and basal parts of the 14th internode with length of 15-17 cm were collected, respectively. Every part had at least 3 biological replicates. Samples for isolating total RNA were stored in liquid nitrogen. (Lines 549-552)

To observe the situation of cell wall at the upper, middle, and basal part of the internodes, we measure the lignin and cellulose content of the three parts. Every part had at least 3 biological replicates. (Lines 559-561)

Total RNA of the internodes was isolated by an RNA extraction kit (Tiangen, Beijing, China). Every part had at least 3 biological replicates. (Lines 574-575)

Reviewer 2 Report

The manuscript titled “Genome-wide identification, expansion and evolution analysis of homeobox gene family reveals TALE gene important for secondary cell wall biosynthesis in moso bamboo” showed the function and evolution of TALE gene family in Bamboo. Manuscript is perfectly planned and conducted, and hence should be accepted for publication without any changes.

Author Response

We thank the reviewer for his/her recognition of our work.